

# Automatic identification and morphological comparison of bivalve and brachiopod fossils based on deep learning

Jiarui Sun[1], Xiaokang Liu[1,2], Yunfei Huang[3], Fengyu Wang[1], Yongfang Sun[1], Jing Chen[4], Daoliang Chu[1] and Haijun Song[1]

[1] State Key Laboratory of Biogeology and Environmental Geology, School of Earth Sciences, China University of Geosciences, Wuhan, Hubei, China
[2] Department of Biology, University of Fribourg, Fribourg, Switzerland
[3] School of Geosciences, Yangtze University, Wuhan, Hubei, China
[4] Yifu Museum, China University of Geosciences, Wuhan, Hubei, China

Corresponding author
Haijun Song, haijunsong@cug.edu.cn

## ABSTRACT

Fossil identification is an essential and fundamental task for conducting palaeontological research. Because the manual identification of fossils requires extensive experience and is time-consuming, automatic identification methods are proposed. However, these studies are limited to a few or dozens of species, which is hardly adequate for the needs of research. This study enabled the automatic identification of hundreds of species based on a newly established fossil dataset. An available "bivalve and brachiopod fossil image dataset" (BBFID, containing >16,000 "image-label" data pairs, taxonomic determination completed) was created. The bivalves and brachiopods contained in BBFID are closely related in morphology, ecology and evolution that have long attracted the interest of researchers. We achieved >80% identification accuracy at 22 genera and ~64% accuracy at 343 species using EfficientNetV2s architecture. The intermediate output of the model was extracted and downscaled to obtain the morphological feature space of fossils using t-distributed stochastic neighbor embedding (t-SNE). We found a distinctive boundary between the morphological feature points of bivalves and brachiopods in fossil morphological feature distribution maps. This study provides a possible method for studying the morphological evolution of fossil clades using computer vision in the future.

## INTRODUCTION

Fossil identification is a fundamental task in palaeontological research and has a wide range of applications, including biostratigraphic dating (*Yin et al., 2001*; *Gradstein et al., 2012*), biological evolution (*Alroy et al., 2008*; *Fan et al., 2020*; *Song et al., 2021*), palaeoenvironmental reconstruction (*Flügel & Munnecke, 2010*; *Scotese et al., 2021*), and palaeoelevational estimation (*Su et al., 2019*). Because taxonomic identification requires a

large amount of prior knowledge as a foundation, researchers need several years of training to accumulate enough experience to ensure the reliability of identification. However, the actual identification process still takes considerable time and is susceptible to subjective factors. The identification accuracy of some genera is even lower than 80% (*Hsiang et al., 2019*). In many fields of palaeontology, deep convolutional neural network (DCNN) has a significant advantage over humans, such as the identification of cut and trampling marks on bones (*Byeon et al., 2019*), the discrimination of dinosaur tracks (*Lallensack, Romilio & Falkingham, 2022*), and the quantification of plant mimesis (*Fan et al., 2022*). To reduce the workload and work difficulty for researchers, automatic fossil identification methods relying on machine learning have been proposed extensively in recent years, among which models using convolutional neural networks (CNNs) (*e.g.*, VGG-16 (*Simonyan & Zisserman, 2014*), Inception-ResNet (*Szegedy et al., 2017*), GoogLeNet (*Szegedy et al., 2015*), *etc.*) have achieved good results (*Dionisio et al., 2020*; *Liu & Song, 2020*; *Liu et al., 2023*; *Niu & Xu, 2022*; *Wang et al., 2022*; *Ho et al., 2023*). Other supervised (*e.g.*, Naïve Bayes) algorithms also achieved ≥70% accuracy in ammonoid species identification (*Foxon, 2021*). This method can assist researchers in fossil identification, reduce the work stress of non-palaeontologists, and enable better identification and application of fossil materials in research. Furthermore, for identifying poorly preserved fossils, neural networks still maintain high identification accuracy (*Bourel et al., 2020*). Neural network in fossil identification is still at an early stage of development, and professional palaeontologists have advantages that such models do not. The ability to take into account complex contextual information is one of them. But in the face of the reality that there are fewer and fewer experts on taxonomy, neural network can provide a useful aid to manual identification rather than replace it (*De Baets, 2021*).  It is still worth studying, and as more training data are available, the reliability and applicability of models will become better.

The training of automatic taxonomy identification models (ATIM) requires a large dataset of labelled fossil images (*Liu et al., 2023*). In non-specialist identification tasks, machine-learning datasets contain millions of images (*e.g.*, ImageNet dataset), which far exceed fossil datasets. The lack of high-resolution (genus-level) fossil labels in the field of palaeontology is mainly due to the tedious and time-consuming process of dataset building. Machine learning has now achieved good results in fossil identification (above the genus level). *Liu & Song (2020)* achieved 95% accuracy for 22 fossil and abiotic grain groups during carbonate microfacies analysis. While 90% accuracy was achieved in the automatic identification of 50 fossil clades relying on web crawlers (*Liu et al., 2023*), genus- and species-level automatic identification focused mainly on a few taxa (mostly < 10). *Dionisio et al. (2020)* performed automatic identification of 9 radiolarian genera, obtaining 91.85% accuracy. *Niu & Xu (2022)* performed automatic identification of fossils covering 113 graptolite species or subspecies. However, similar studies targeting a large number of taxa are less common (Fig. 1, details of the relevant studies can be found in Appendix S1). In practice, it is common to identify a large number of fossil categories. However, current automatic identification studies are limited to a few or dozens of taxa, which is hardly adequate for the needs of research. There is a gap in automatic fossil identification studies for hundreds of taxa. This study provides new practice in this field based on a newly
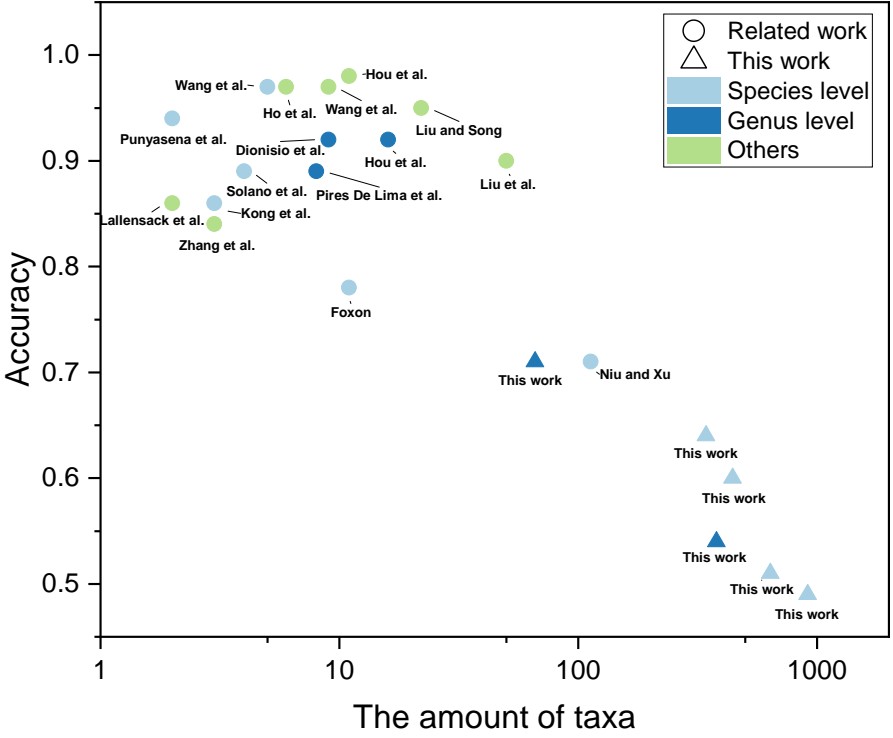

**Figure 1** Number of taxa and accuracy for automatic fossil identification studies based on deep learning (*Punyasena et al., 2012*; *Kong, Punyasena & Fowlkes, 2016*; *Solano, Gasmen & Marquez, 2018*; *Dionisio et al., 2020*; *Hou et al., 2020*; *Liu & Song, 2020*; *Pires De Lima et al., 2020*; *Zhang et al., 2020*; *Foxon, 2021*; *Lallensack, Romilio & Falkingham, 2022*; *Niu & Xu, 2022*; *Wang et al., 2022*; *Ho et al., 2023*; *Hou et al., 2023*; *Liu et al., 2023*; *Wang et al., 2023*).

established fossil dataset. In addition, previous studies all focus on the same fossil clade (*e.g.*, radiolarians, brachiopods, *etc.*). It is unclear whether fossils in different phyla can achieve automatic fossil identification.

Brachiopods and bivalves are the two most common invertebrate clades in the Phanerozoic (*Sepkoski, 1981*; *Clapham et al., 2006*; *Benton & Harper, 2020*). Brachiopods are the dominant fossil animals of the Paleozoic, but their diversity is now far less than that of bivalves (*Thayer, 1986*). The start of this transition occurred at the Permian-Triassic mass extinction (PTME), when the marine benthic faunas changed from brachiopod-dominated Paleozoic evolutionary fauna to mollusk-dominated modern evolutionary fauna (*Fraiser & Bottjer, 2007*; *Dai et al., 2023*). The reasons for the dominance of bivalves over brachiopods have long attracted the attention of palaeontologists (*Ballanti, Tullis & Ward, 2012*; *Payne et al., 2014*). The similarities and differences between them in morphology and physiological mechanisms may be an important perspective. Whether bivalves and brachiopods influenced each other evolutionarily is a controversial issue, also known as "ships that pass in the night" (*Gould & Calloway, 1980*; *Fraiser & Bottjer, 2007*; *Liow, Reitan & Harnik, 2015*). Bivalves feed more efficiently at high algal concentrations than articulate brachiopods, which is thought to be the reason for the physiological perspective

(*Rhodes & Thompson, 1993*). Morphologically, the prosperity of the bivalves after PTME cannot be attributed to their morphological innovations (*Fraiser & Bottjer, 2007*), while bivalves suppressed brachiopod evolution (*Liow, Reitan & Harnik, 2015*). However, they still have certain similarities, for instance they both tend to become smaller under heat stress (*Piazza, Ullmann & Aberhan, 2020*). The close relationship and significant differences between them have attracted researchers' interest, and conducting morphological studies is the first step. However, the similar morphological features between them have caused problems for researchers to identify them accurately.

Automatic identification of brachiopods has been carried out previously. *Wang et al. (2022)* used the transposed convolutional neural network to realize the automatic identification of fossils with a relatively small dataset and they achieved 97% identification accuracy for five brachiopod species based on 630 training images. In this study, we enabled the automatic identification of hundreds of taxa (bivalve and brachiopod) based on a newly established fossil dataset. We built a "bivalve and brachiopod fossil image dataset" (BBFID) (16,596 labelled fossil images covering 870 genera and 2033 species) for the first time by collecting and sorting a large amount of published literature. We built ATIMs using transfer learning in VGG-16 (*Simonyan & Zisserman, 2014*), Inception-ResNet-v2 (*Szegedy et al., 2017*), and EfficientNetV2s (*Tan & Le, 2021*) architectures, which have performed well in general identifications. Furthermore, we extracted the process outputs of the model as fossil features and downscaled them to two-dimensional data using t-SNE (*Van Der Maaten & Hinton, 2008*). Plotting them in a two-dimensional space is an effective way to compare morphological differences between bivalves and brachiopods.

## MATERIALS AND DATA

The BBFID used for training ATIMs contains bivalve-part (BBFID-1) and brachiopod-part (BBFID-2), all collected from published literature and monographs (see Appendix S6). Detailed data on the number of each taxon are given in Appendix S6 (Tables S1 and S2). This study collected fossil images from publications that were of diverse origin. This makes use of the large amount of data that already exists and allows for better use of data from previous studies.

We used Adobe Acrobat Pro DC to capture accurately named bivalve and brachiopod fossil images from the collected literature and saved them as BMP, JPG, or PNG images to minimize the quality loss of the images. These fossils are Carboniferous (~0.08%), Permian (majority, ~39.5%), Triassic (majority, ~58.9%), Jurassic (~1.4%), and Quaternary (~0.03%) in age. Permian and Triassic fossils make up the vast majority (~98.5%) of the dataset. Their overlapping occurrences, having undergone the same geological events, are of great importance in fossil identification and in studies of morphological evolution. Those that could not be saved due to the encryption of PDF files in the literature were screenshotted as PNG files using Snipaste. The majority of images collected from plates are single animal images, and the effect of plate numbering was avoided as much as possible.

We obtained more than 16,000 fossil images from 188 publications and performed data cleaning. The contribution of each publication to the dataset is given in Appendix S2. During the data collection stage, we collected as many fossil images as possible. These

images were taken at any viewpoint and in any orientation. Different views of the same specimen were treated as different instances and labelled separately. To ensure the reliability of the dataset, we checked the bivalve and brachiopod images and corresponding labels. Because the taxonomic system of bivalves and brachiopods is continuously improving (*Konopleva et al., 2017*; *Sulser et al., 2010*), we categorized the genera whose taxonomic names and positions had been changed in the literature. Additionally, we removed poorly preserved fossil images, which contain two cases. The first case is images with uncertain taxonomic names. The other discarded images are obtained from scanned published documents (mostly monographs published in the last century) that are poorly pixelated and difficult to identify even for palaeontologists. In both cases, the ambiguous images are discarded based on whether the experts can distinguish the fossils or not. There is no filtering based on deep learning preference, so this operation does not affect the utility of the deep learning method.

Our dataset was randomly divided into the training set (60%), validation set (20%), and test set (20%) to train, tune, and test the model. Such a distribution is intended for the test set to cover a sufficient number of taxa to make the accuracy more reliable. Because the validation set is used as a reference for the tuning process, the identification accuracy of this part may have artificial bias and is not universally meaningful. Thus, the final accuracy was calculated using a separate test set to evaluate model performance.

The final BBFID contains 870 genera, with 16,596 sets of "image-label" data pairs. All images have genus labels, with 14,185 items having higher-resolution species labels. BBFID-1 contains 379 genera and 889 species, with 8,144 sets of image-label data pairs. BBFID-2 contains 491 genera and 1,144 species, with 8,452 sets of data pairs. A total of about 2,300 genera of bivalves have been described to date, and 1,700 genera of brachiopods have been described (*Pitrat & Moore, 1965*; *Nevesskaja, 2003*). BBFID covers 16.4% and 28.8% of the described bivalve and brachiopod genus-level classifications, respectively. Genus distributions of BBFID are shown in Fig. 2. The BBFID-1 dataset consists of ~85% black and white images, and the rest are in colour. In situ photographs of fossils (images with rocks in the background) occupy ~25% of BBFID-1, while all other fossil photographs have plain white/black backgrounds. The BBFID-2 dataset consists of ~95% black and white images, and the rest are in colour. In situ photographs of fossils (images with rocks in the background) occupy ~1% (61 images) of BBFID-2, while all other fossil photographs have plain white/black backgrounds. The BBFID-2 (brachiopod dataset) has more plain white/black background photographs, because brachiopod fossils are more robust and easier to preserve intact than bivalve fossils. Therefore the former can be imaged to obtain more complete pictures of the fossils without the rocky background.

To meet the requirements of machine learning, each taxon should have at least three items. Therefore, we chose the categories with >2 items of BBFID to perform the model training, which contains 16,389 sets of "image-label" data pairs. BBFID contains images of the whole shells and detailed images. Detailed images refer to all non-full shell face images as well as photographs not in front view, such as structures of fossils. Of the selected

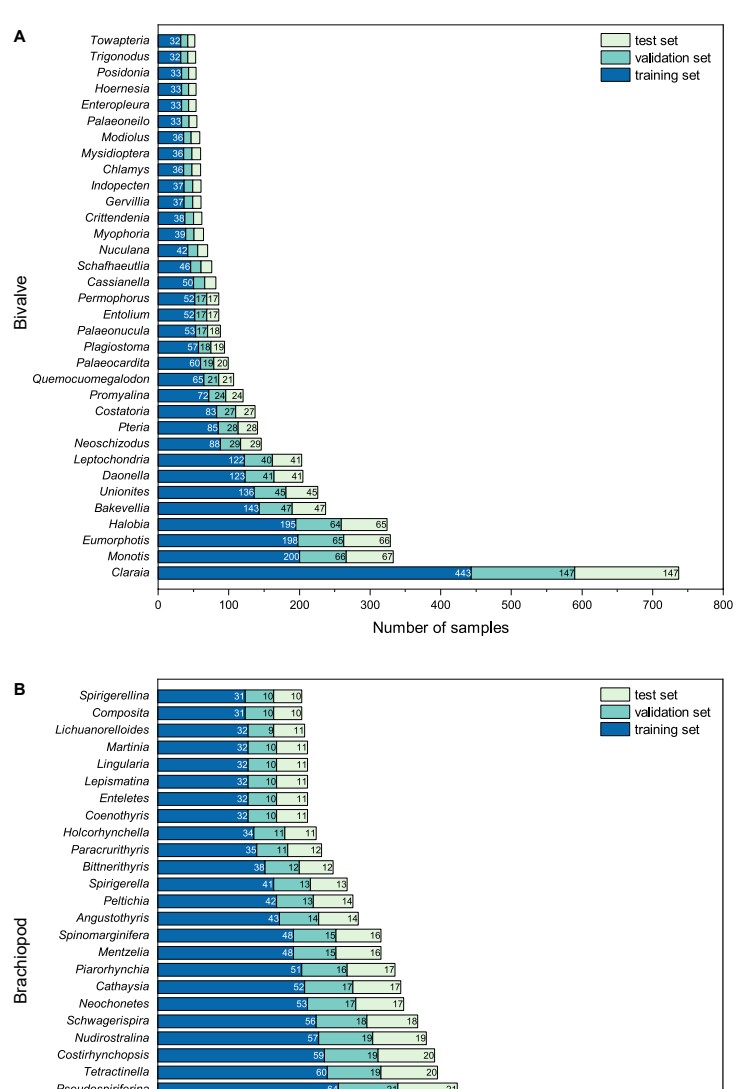

**Figure 2** Number of samples for each taxon at the genus level in (A) BBFID-1 and (B) BBFID-2 (scale B) and the distribution in subsets.

images, detailed images occupy ~18% in BBFID-1, ~40% in BBFID-2, and ~29% in the overall BBFID. The number of detailed images (the categories with >2 items) and the exact number of detailed images in each dataset (training set, validation set, and test set) of the common genera are available in Appendix S6 (Tables S1 and S2).

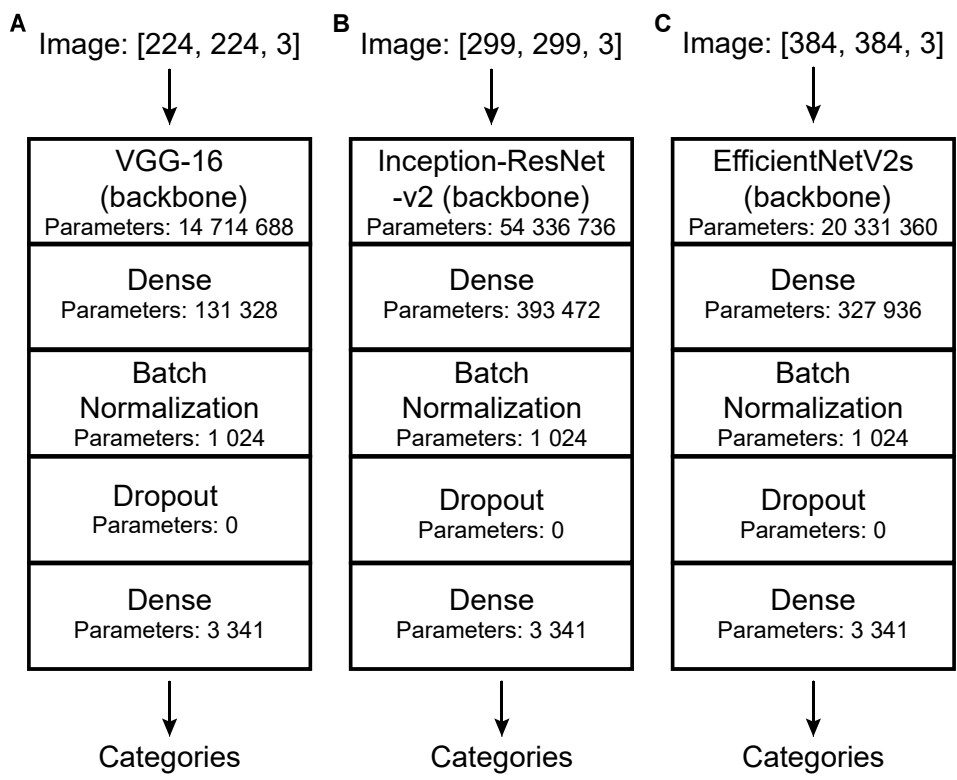

**Figure 3** **DCNN architectures used in this study.** Automatic identification model architectures of (A), (B), and (C) are modified from VGG-16 (*Simonyan & Zisserman, 2014*), Inception-ResNet-v2 (*Szegedy et al., 2017*), and EfficientNetV2s (*Tan & Le, 2021*), respectively.

## METHODS

### Convolutional neural network

Convolutional neural networks (CNNs) perform well in general recognition and have been used in the automatic identification of palaeontological fossils (*Dionisio et al., 2020*; *Liu & Song, 2020*; *Kiel, 2021*; *Liu et al., 2023*; *Niu & Xu, 2022*; *Wang et al., 2022*; *Ho et al., 2023*). In this study, three pre-trained models of convolutional neural networks with good classification performance on the ImageNet dataset (*Deng et al., 2009*) namely VGG-16 (*Simonyan & Zisserman, 2014*), Inception-ResNet-v2 (*Szegedy et al., 2017*), and EfficientNetV2s (*Tan & Le, 2021*) were selected and suitably modified (Fig. 3). VGG-16 and Inception-ResNet-v2 have been proven to automatically identify fossils and perform well (*Hsiang et al., 2019*; *Liu et al., 2023*). We retained their main architecture, removed the top softmax layer and/or fully connected layer depending on fossil categories, and added a fully connected layer (with 256 output and Relu activation function), batch normalization layer (*Ioffe & Szegedy, 2015*), dropout layer (with rate = 0.2), and fully connected layer (with output as fossil categories) (Fig. 3).

In fossil identification, CNNs first decode the fossil images to obtain the tensor that can be operated, and the model operates on these values to establish the correspondence

between the image data and the fossil name. CNNs use convolutional operations to process image data and gradient descent to minimize the loss function to train the model (*Lecun et al., 1998*). The neural network can be divided into multiple network layers. More specifically, the convolutional, pooling, and fully connected layers play a crucial role in the automatic identification process. The convolutional layers transform an image by sweeping a kernel over each pixel and performing a mathematical operation. The pooling layer reduces the amount of computation, making the model easier to train (*Giusti et al., 2013*). The fully connected layer and activation function (Relu) fit the correspondence between fossils and labels (*Nair & Hinton, 2010*) and output the predicted labels and probabilities we need at the top layer.

VGG-16 is a classic DCNN proposed by *Simonyan & Zisserman (2014)*, which uses 16 layers and $3 \times 3$ convolutional kernels (convolution filters) to achieve good performance. Then, *He et al. (2016)* proposed a new residual connectivity method and applied it to Inception-ResNet-v2, which makes the network easier to optimize and allows the use of a deeper network to improve performance. EfficientNetV2 is currently a more advanced open-source image classification model using the training-aware neural architecture search and scaling method to improve training speed and parameter efficiency (*Tan & Le, 2021*).

## Data preprocess

Deep learning models have requirements for input data size. However, the images in our dataset were of different sizes and the labels were also inappropriate for model training. Thus, data needed to be preprocessed. To match the model's requirement, all images were resized to a uniform size (slightly different depending on the model: VGG-16, 224*224; Inception-ResNet-v2, 299*299; EfficientNetV2s, 384*384.). Further details are shown in Fig. 3. To improve their generalization ability, we randomly adjusted the image (training set and validation set) brightness (within $\pm$ 0.5) and contrast (within 0 to +10) (*Simonyan & Zisserman, 2014*; *Szegedy et al., 2015*; *He et al., 2016*; *Liu & Song, 2020*). In addition, the images were normalized and standardized [all images were processed using the following equation: $x\_new = (x\text{-mean})/std$, mean $= (0.485, 0.456, 0.406)$, std $= (0.229, 0.224, 0.225)$. Mean and std are empirical values, which are calculated from a large number of images]. We used discrete one-hot coding for image labels. Finally, a one-to-one correspondence between the images and the labels was established, and we obtained the processed machine-learning dataset.

## Training methodology

Achieving high accuracy in multiclass fossil identification using neural networks requires a large dataset as a basis. Although we built the bivalve and brachiopod dataset manually, it was still insufficient to train a model with random initialization of parameters to converge and achieve the best results. Therefore, we applied transfer learning in the model training process, an effective way to train a model on a small dataset (*Tan et al., 2018*; *Brodzicki et al., 2020*; *Koeshidayatullah et al., 2020*). Transfer learning uses parameters trained by general identification tasks for initialization to accelerate the convergence of the new model. It is feasible to use this to reuse the general identification model parameters for

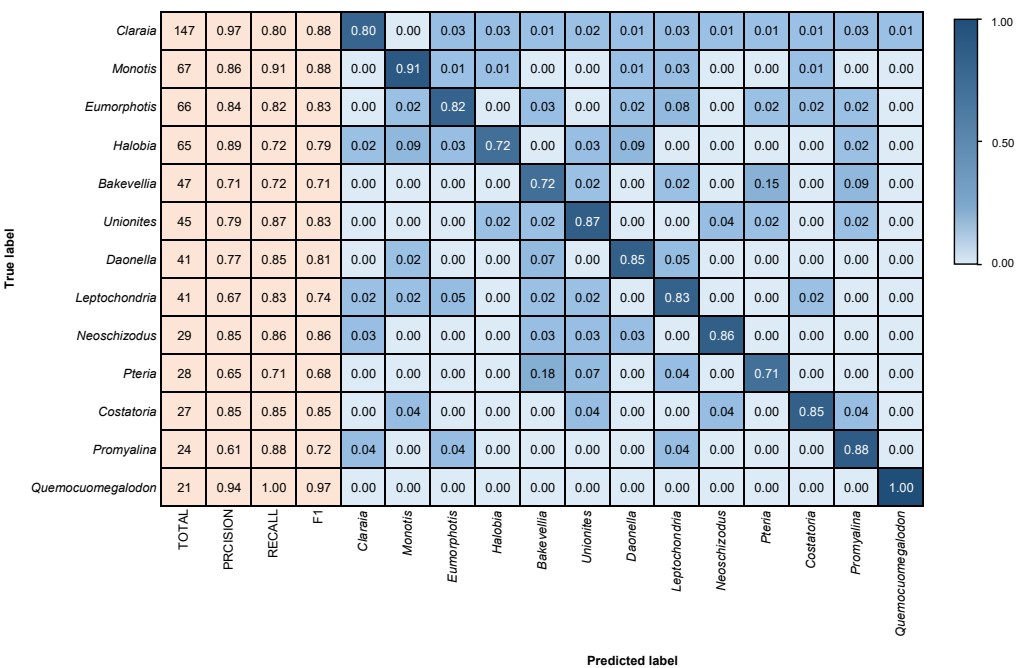

**Figure 4 Confusion matrix and evaluation metrics of models trained by BBFID-1 (scale A) on genus mode.** The horizontal axis is the predicted label, and the vertical axis is the true label. Colors and values represent the proportion of the corresponding taxon identified as the predicted label taxon.

palaeontological fossil identifications (*Pires De Lima et al., 2020*). This is why we only envision applying this method to the automatic identification of common fossils, while fossils with too few specimens will still need to rely on palaeontologists.

In this study, each model was loaded with pre-trained parameters that were originally trained on ImageNet. This method greatly reduces the amount of data required for automatic identification, greatly expanding their application scenarios.

We coded in Python and relied on the Tensorflow scientific computing library (*Abadi et al., 2016*) to train the model. The training process was performed using the Adam optimizer (*Kingma & Ba, 2014*). The loss function uses the categorical cross-entropy loss function (*Botev et al., 2013*), and the accuracy is used as an evaluation metric during training. The final model performance is presented using a confusion matrix and F1 score (Figs. 4–6). The confusion matrix contains recall and precision, which represent two perspectives of identification performance. Recall represents the proportion of "items correctly identified as a specific taxon" to "the total items belong to that taxon". Precision represents the proportion of "items correctly identified as a specific taxon" to "the total items identified as that taxon". The F1 score is the harmonic mean of the recall and precision, which can represent both false positives and false negatives (*Sarkar, Bali & Ghosh, 2018*).

To facilitate training, the learning rate is adjusted with validation loss in training. If the decrease in "validation loss" is less than 0.0001 for 5 epochs, the learning rate will be halved using "callbacks.ReduceLROnPlateau()" function. Additionally, to prevent overfitting,
EarlyStopping (a method to stop training when the model performs optimally) was set to ensure the good performance of the model in the test set (*Ying, 2019*). During the training process, the model saves architecture and parameters with the highest accuracy in the validation set in real-time for rapid deployment in subsequent applications. Because BBFID contains both the genus tags and species tags, we set the model to the genus mode (only read the genus tag) and species mode (read both genus tag and species tag) during model training and testing. Because of dataset size, the model's architecture and hyperparameters significantly affect its performance; thus, we trained models and compared their performance under different scenarios (Table 1).

We chose the different sizes of the datasets to train models according to the taxonomic levels. At the genus-level, we set three scales to explore model performance using different volumes of datasets. These three scales are the number of each genus >100 images (scale A), >50 images (scale B), and >10 images (scale C) (Table 2). Among them, scale B/C contains all genera with more than 50/10 pictures, the same for other scales. The numbers of taxa in BBFID-1 are 13 (scale A), 34 (scale B), and 156 (scale C), whereas the numbers of items in BBFID-2 are 9 (scale A), 32 (scale B), and 223 (scale C). They display a clear gradient to match our research needs. For the selection of data adequacy (*i.e.,* data scale) at the species level, we selected scale B (number of each species >50 images) and scale C (number of each species >10 images) for training and testing, according to the performance of the genus mode. Furthermore, we also tried two larger scales: scale D (number of each species >8 images) and scale E (number of each species >6 images). There are four gradients in total to find the range that covers more genera with guaranteed accuracy. In addition, for BBFID, we added two larger scales (the number of each taxon >4 images and >2 images) to explore the model performance in small datasets. As mentioned earlier, all data (scales A, B, C, D, and E) were randomly divided into the training set, validation set, and test set in the ratios of 60%, 20%, and 20%, which is the ideal situation. In order to try a larger data scale, we discarded the requirement that the validation set cover all species. Therefore, the number of single-taxon images >2 was the maximum data size we could try, because all taxonomic units shall be covered in the training set and test set.

Model architecture plays a pivotal role in models. Thus, we used BBFID-1 (scale A) to test model identification accuracy at the genus level under three different model architectures (*i.e.,* VGG-16, Inception-ResNet-v2, and EfficientNetV2s). Subsequently, the best architecture was selected to build ATIM, trained and tested using different scales of BBFID-1, BBFID-2, and BBFID, respectively, to obtain the corresponding model performance (Table 2).

Considering that a particular identification model cannot identify arbitrary fossil taxa, it is necessary to establish a method for measuring the applicability of the model. We divide the entire BBFID into "applicable" and "inapplicable". Anything in the training set is considered "applicable" and anything not in training set is considered "inapplicable". Binary classification training based on Inception-ResNet-v2 was performed and the "Applicability Model" (AM) was obtained. Users can use the AM to determine the applicability.

**Table 1   Identification accuracy training on BBFID-1 (scale A) at the genus level with different model architectures and hyperparameters.** Architectures in this table are shown in Fig. 3. "Trainable layers of functional layers" represents the size of the parameters that can be trained. "None" means that all layers of the backbone are frozen and the parameters involved in these layers cannot be trained. These parameters maintain the values at the time of model initialization. "Half layers" means that half of the backbone layer parameters are frozen, while "All layers" means that all parameters of this model are not frozen and can be updated during the training process. This setting has an impact on both the model training process and the model performance.

| Order | Backbone | Batch size | Trainable layers of functional layers | Reduce LR on plateau | Epochs | Max. training accuracy | Min. training loss | Max. validation accuracy | Min. validation loss | Test accuracy | Test loss |
|---|---|---|---|---|---|---|---|---|---|---|---|
| 1 | VGG-16 | 32 | None | Yes | 50 | 0.8648 | 0.4212 | 0.6444 | 1.1440 | 0.6281 | 1.2512 |
| 2 | VGG-16 | 32 | Half layers | Yes | 40 | 0.9959 | 0.0181 | 0.7515 | 0.9126 | 0.7330 | 0.8444 |
| 3 | VGG-16 | 32 | All layers | Yes | 50 | 0.7670 | 0.6080 | 0.5698 | 1.3465 | 0.5386 | 1.4802 |
| 4 | VGG-16 | 32 | All layers | No | 36 | 0.3609 | 1.8002 | 0.3338 | 2.0523 | 0.0957 | 3.0871 |
| 5 | Inception-ResNet-v2 | 8 | None | Yes | 50 | 0.3236 | 1.9945 | 0.3385 | 2.0345 | 0.3225 | 2.1000 |
| 6 | Inception-ResNet-v2 | 8 | Half layers | Yes | 50 | 0.7363 | 0.7163 | 0.5263 | 1.4931 | 0.4877 | 1.5584 |
| 7 | Inception-ResNet-v2 | 8 | All layers | Yes | 46 | 0.9959 | 0.0216 | 0.7934 | 1.2041 | 0.7778 | 2.5044 |
| 8 | Inception-ResNet-v2 | 8 | All layers | No | 46 | 0.9805 | 0.0602 | 0.7981 | 0.8178 | 0.6590 | 1.2590 |
| 9 | EfficientNetV2s | 8 | None | Yes | 50 | 0.5693 | 1.2799 | 0.5419 | 1.4210 | 0.4923 | 1.5424 |
| 10 | EfficientNetV2s | 8 | Half layers | Yes | 50 | 0.9708 | 0.1013 | 0.7624 | 0.8314 | 0.7515 | 0.8633 |
| 11 | EfficientNetV2s | 8 | All layers | Yes | 44 | 0.9959 | 0.0139 | 0.8338 | 0.6130 | 0.8302 | 0.6807 |
| 12 | EfficientNetV2s | 8 | All layers | No | 37 | 0.9825 | 0.0578 | 0.8136 | 0.7905 | 0.7886 | 0.8122 |

**Table 2** **Model performance using BBFID-1, BBFID-2 and BBFID in EfficientNetV2s architecture.** Learning rate starts from 1e−4 and the epoch is limited to less than 51. Test accuracy/Test loss means the accuracy / loss of the saved model.

| Order | MODE | Dataset | Scale | > x images each taxon | Number of categories | Learning rate in the end | Epochs | Max. training accuracy | Min. training loss | Max. validation accuracy | Min. validation loss | Last epoch test accuracy | Last epoch test loss | Test accuracy | Test loss |
|---|---|---|---|---|---|---|---|---|---|---|---|---|---|---|---|
| 13 | Genus | BBFID-1 | C | 10 | 156 | 1.25E−05 | 49 | 0.9972 | 0.0080 | 0.5990 | 1.8758 | 0.5848 | 1.9234 | 0.5834 | 1.9320 |
| 14 | Genus | BBFID-1 | B | 50 | 34 | 1.25E−05 | 34 | 0.9939 | 0.0281 | 0.7185 | 1.1308 | 0.6916 | 1.1420 | 0.7173 | 1.1142 |
| 15 | Genus | BBFID-1 | A | 100 | 13 | 5.00E−05 | 29 | 0.9866 | 0.0446 | 0.8090 | 0.6661 | 0.8256 | 0.6719 | 0.8210 | 0.6650 |
| 16 | Genus | BBFID-2 | C | 10 | 223 | 1.00E−04 | 22 | 0.9908 | 0.0848 | 0.5320 | 2.1067 | 0.4919 | 2.2493 | 0.5004 | 2.2021 |
| 17 | Genus | BBFID-2 | B | 50 | 32 | 5.00E−05 | 21 | 0.9929 | 0.0483 | 0.7370 | 0.9765 | 0.7170 | 1.0273 | 0.7135 | 1.0625 |
| 18 | Genus | BBFID-2 | A | 100 | 9 | 5.00E−05 | 25 | 0.9878 | 0.0486 | 0.8636 | 0.5007 | 0.8259 | 0.5409 | 0.8543 | 0.4904 |
| 19 | Genus | BBFID | C | 10 | 379 | 2.50E−05 | 35 | 0.9974 | 0.0134 | 0.5567 | 2.0772 | 0.5353 | 2.2279 | 0.5371 | 2.2333 |
| 20 | Genus | BBFID | B | 50 | 66 | 2.50E−05 | 27 | 0.9933 | 0.0299 | 0.7335 | 1.1080 | 0.7192 | 1.1866 | 0.7066 | 1.2000 |
| 21 | Genus | BBFID | / | 60 | 47 | 1.25E−05 | 34 | 0.9961 | 0.0177 | 0.7538 | 1.0721 | 0.7742 | 0.8506 | 0.7626 | 0.8921 |
| 22 | Genus | BBFID | A | 100 | 22 | 5.00E−05 | 26 | 0.9907 | 0.0335 | 0.8261 | 0.6590 | 0.8190 | 0.6615 | 0.8145 | 0.6759 |
| 23 | Species | BBFID-1 | E | 6 | 241 | 5.00E−05 | 31 | 0.9949 | 0.0345 | 0.6117 | 1.8168 | 0.5971 | 1.9054 | 0.6080 | 1.9233 |
| 24 | Species | BBFID-1 | D | 8 | 179 | 1.00E−04 | 28 | 0.9938 | 0.0645 | 0.6251 | 1.6484 | 0.5810 | 1.8759 | 0.6299 | 1.6987 |
| 25 | Species | BBFID-1 | C | 10 | 148 | 2.50E−05 | 32 | 0.9975 | 0.0289 | 0.6629 | 1.4035 | 0.6642 | 1.4147 | 0.6790 | 1.4161 |
| 26 | Species | BBFID-1 | B | 50 | 8 | 5.00E−05 | 27 | 0.9871 | 0.0789 | 0.7460 | 0.7560 | 0.7984 | 0.7489 | 0.8140 | 0.6747 |
| 27 | Species | BBFID-2 | E | 6 | 396 | 1.00E−04 | 23 | 0.9950 | 0.0726 | 0.5128 | 2.3015 | 0.4677 | 2.5728 | 0.4813 | 2.5160 |
| 28 | Species | BBFID-2 | D | 8 | 265 | 1.00E−04 | 28 | 0.9983 | 0.0492 | 0.5590 | 1.9957 | 0.5411 | 2.0768 | 0.5349 | 2.1075 |
| 29 | Species | BBFID-2 | C | 10 | 195 | 1.00E−04 | 25 | 0.9969 | 0.0647 | 0.6162 | 1.6714 | 0.5540 | 1.9768 | 0.5791 | 1.8711 |
| 30 | Species | BBFID-2 | B | 50 | 8 | 5.00E−05 | 24 | 0.9968 | 0.0472 | 0.9494 | 0.1308 | 0.9615 | 0.1806 | 0.9519 | 0.1610 |
| 31 | Species | BBFID | / | 2 | 1436 | 5.00E−05 | 41 | 0.9956 | 0.0271 | 0.4975 | 2.4540 | 0.4274 | 2.8980 | 0.4330 | 2.9233 |
| 32 | Species | BBFID | / | 4 | 914 | 1.00E−04 | 28 | 0.9920 | 0.0758 | 0.4958 | 2.4228 | 0.4707 | 2.5650 | 0.4899 | 2.5005 |
| 33 | Species | BBFID | E | 6 | 637 | 1.00E−04 | 25 | 0.9934 | 0.0677 | 0.5521 | 2.0340 | 0.5067 | 2.3438 | 0.5142 | 2.2276 |
| 34 | Species | BBFID | D | 8 | 444 | 5.00E−05 | 26 | 0.9975 | 0.0291 | 0.6148 | 1.6785 | 0.5752 | 1.8458 | 0.5957 | 1.8470 |
| 35 | Species | BBFID | C | 10 | 343 | 2.50E−05 | 34 | 0.9991 | 0.0143 | 0.6472 | 1.5119 | 0.6476 | 1.4888 | 0.6397 | 1.5602 |
| 36 | Species | BBFID | B | 50 | 16 | 1.00E−04 | 23 | 0.9787 | 0.1037 | 0.8399 | 0.5760 | 0.8283 | 0.5472 | 0.8283 | 0.5487 |

## Dimensionality reduction method

In this study, we employed a downscaling method of t-SNE that uses a probability measure of similarity and expresses probabilities as spatial distances (*Van Der Maaten & Hinton, 2008*). To compare fossil morphology, we extracted the output of the last maximum pooling layer as fossil features and downscaled the high-dimensional data of fossil features to a two-dimensional plane using t-SNE. Next, we visualized that to analyze easily the morphological differences and similarities between bivalves and brachiopods. The model training and downscaled visualization codes were referenced from some open-source projects (*Liu & Song, 2020*; *Liu et al., 2023*).

## RESULTS

### Model performance between different architectures and hyperparameters

Different architectures perform differently using BBFID-1 (scale A, genus level), with the best performance of 83.02% obtained with the EfficientNetV2s architecture and the corresponding hyperparameters (Table 1). The results of confusion matrix for this identification task are shown in Fig. 4. The identification recalls were >79% for all categories except the genera *Pteria* (0.71, test set: 28 items), *Bakevellia* (0.72, test set: 47 items), and *Halobia* (0.72, test set: 65 items), whereas the accuracies of *Quemocuomegalodon* (1.00, test set: 21 items), and *Monotis* (0.91, test set: 67 items) exceeded 90%.

### Model performance using different data scales

The three model architectures (VGG-16, Inception-ResNet-v2, and EfficientNetV2s) were tested in BBFID-1 and the EfficientNetV2s architecture was found to perform best. We used EfficientNetV2s architecture that performed well on BBFID-1 and corresponding hyperparameters to build other models (genus mode) using different data scales, which performed as expected under different datasets (Table 2). The accuracy of BBFID-1 (scale A) was 82.10%, whereas those of scales B and C were 71.73% and 58.34% respectively, with the loss increasing by decreasing accuracy for all three. The accuracy of BBFID-2 was 85.43%, 71.35%, and 50.04% for the three dataset scales, whereas the identification accuracy of scale A exceeded 85%. Furthermore, in four categories, more than 90% of images were identified correctly (Fig. 5). The accuracy of model training by BBFID was 81.45%, 70.66%, and 53.71% at the three scales, and the performance of each scale was similar to the performance of the corresponding bivalve and brachiopod individual identifications. In species mode, the models also performed similarly (Table 2), with the accuracy of BBFID at scale C (148 categories for bivalves, 195 categories for brachiopods) of more than 60% (see Appendix S3 for confusion matrix and evaluation metrics). The accuracies of Scale D (bivalve 179 categories, brachiopod 265 categories) and scale E (bivalve 241 categories, brachiopod 396 categories) ranged from 51% to 59%. All these models in EfficientNetV2s architecture met the early stopping condition and terminated training before 50 epochs, and the training set accuracy was close to 100% at this point. This indicates that the models completed fitting to the training set. The training process of BBFID (scale A) shows that the model basically converged about 20 epochs (Fig. 7), and

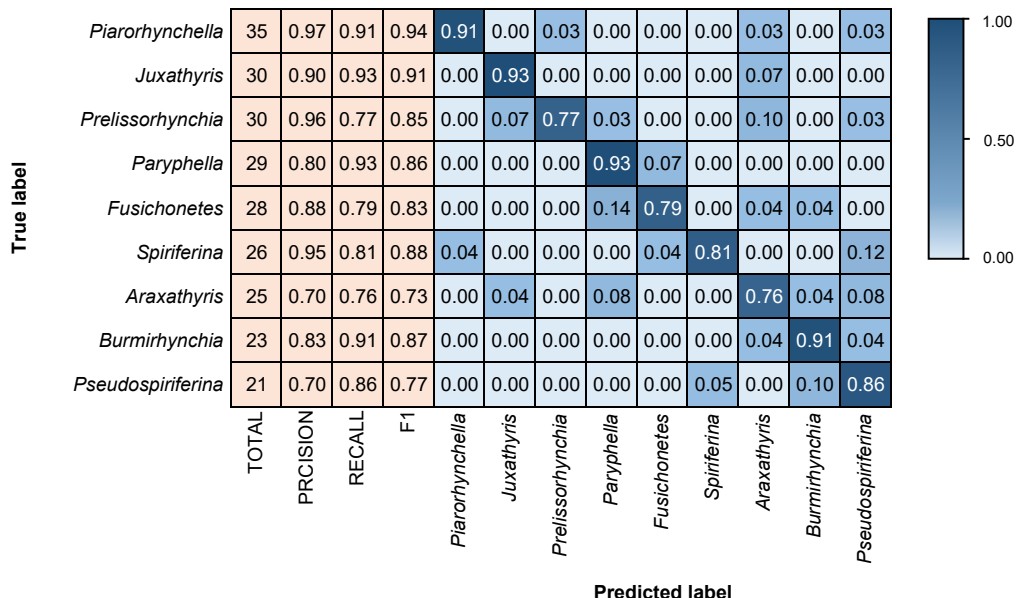

**Figure 5  Confusion matrix and evaluation metrics of models trained by BBFID-2 (scale A) on genus mode.** Colors and values represent the proportion of the corresponding taxon identified as the predicted label taxon.

its training set accuracy finally reached ~100%, while the maximum validation accuracy was over 80% (Table 2).

We extracted the process output from the ATIM (Order 22) and summed the same point data in each dimension to draw a feature map (Fig. 8). We also used the output of the top maximum pooling layer as fossil features and then used t-SNE (*Van Der Maaten & Hinton, 2008*) for dimension reduction, which achieved good results of morphology clustering and comparison (Fig. 9). The classification of each taxon in Fig. 9 is clear, and the t-SNE results are similar between the training set (Fig. 9A) and the validation set and test set (Fig. 9B). However, the individual clusters obtained from the training set are more concentrated and the boundaries between different categories are clearer than the latter due to the training process (Fig. 9). Additional t-SNE calculation for more categories (444 categories, based on Order 34) was also performed (see Appendix S6).

## DISCUSSION

### Identification accuracy

The ranking of automatic identification performance among three architectures trained by BBFID-1 (Table 1) is comparable to general task results (*Simonyan & Zisserman, 2014*; *Szegedy et al., 2017*; *Tan & Le, 2021*), indicating that transfer learning is useful. It is feasible to apply pre-trained parameters of the general model to the ATIM in the field of palaeontology using transfer learning. The identification accuracy (>80%) on genus mode is similar to some previous studies that built upon ResNet architecture (*Romero et al., 2020*). *Romero et al. (2020)* achieved an accuracy of 83.59% using the external morphology
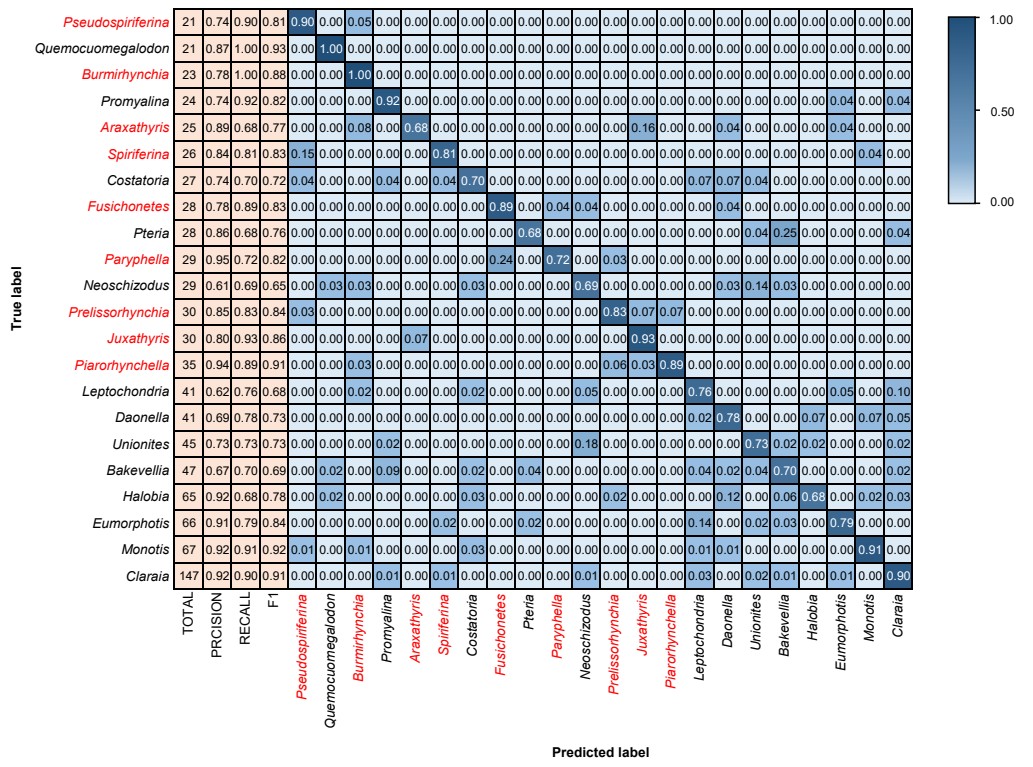

**Figure 6 Confusion matrix and evaluation metrics of models trained by BBFID (scale A) on genus mode.** Colors and values represent the proportion of the corresponding taxon identified as the predicted label taxon. The categories marked in red are brachiopods, and the others are bivalves.

of pollen grains, increasing to 90% with the addition of an internal structure using Airyscan confocal superresolution microscopy. Adding the sequential internal structure of bivalves and brachiopods may be a way to improve identification accuracy.

The images of BBFID come from over 100 publications, whereas the contributions of the publications are not well balanced (*e.g.*, some publications contribute >1,000 images, while some only contribute <100, see Appendix S2). The imbalance in publication contributions was not specifically processed. This is a limitation of the data sources and could potentially impact the model's generalisation ability. The impact level still needs to be explored in future studies. The fossil images used in this study contain pictures of the whole shells and detailed pictures, such as structures of fossils. The detailed images contain different information than the whole shell images. Since no specific labels have been added to the detail images, the identification accuracy was adversely affected by this factor. For the accuracy of different parts of the dataset, the accuracy of the validation set was comparable to that of the test set, but lower than that of the training set. Because the model was trained using the training set, the identification performance was better in this part. However, the data from the validation and test sets were not used to train the models. Accordingly, the results were slightly worse compared with the training set. Furthermore, the validation set was purposefully optimized in the conditioning.

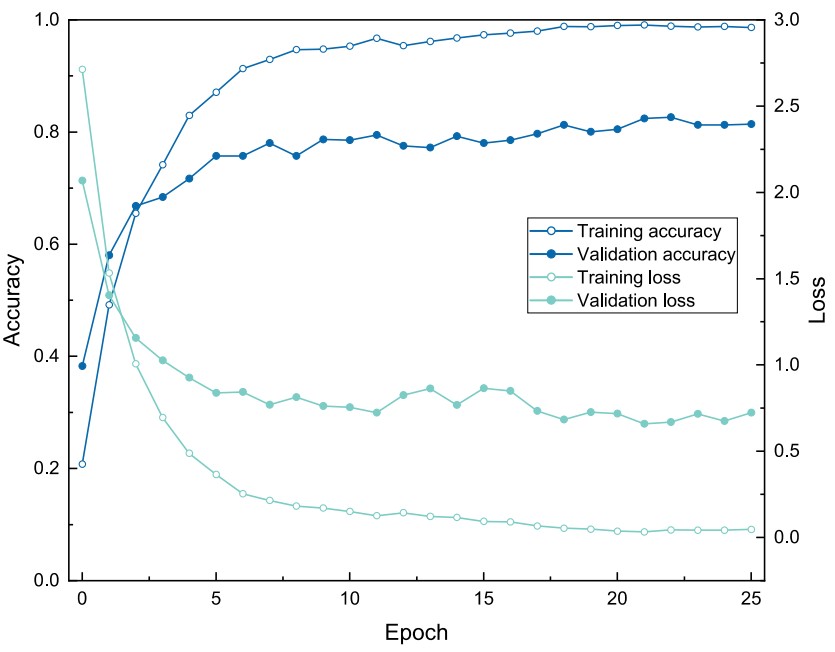

**Figure 7** The training process of ATIM on genus mode using BBFID (scale A) (Order 22).

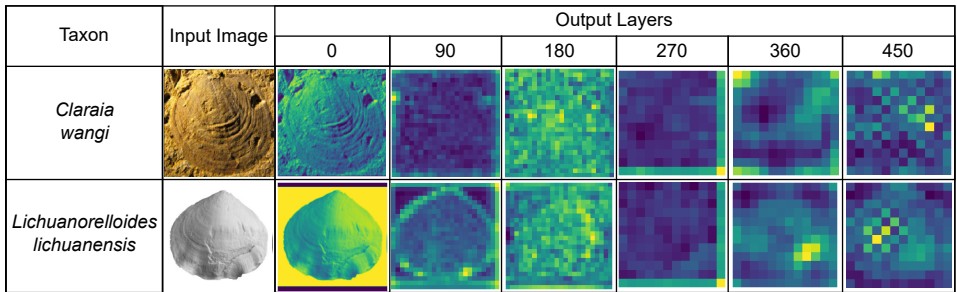

**Figure 8** Feature maps of the bivalve (*Claraia*) and brachiopod (*Lichuanorelloides*) fossils in BBFID, plotted by extracting model (Order 22) intermediate output. Fossil images are from *Huang, Tong & Fraiser (2018)*, and *Wang et al. (2017)*.

The accuracy of the model using selected architecture and parameters (Table 1, Order 11) on genus mode exceeded 80% using BBFID-1 (scale A). In contrast, the accuracy decreases between scale B and scale C stems from the decrease in single taxon images and confusion caused by the categories increase. Nevertheless, the identification accuracy of scale C (156 categories) was still close to 60%. In addition, the model based on BBFID-2 achieved similar accuracy to the model based on BBFID-1 at all scales. The identification accuracy at scale A exceeded 80%, which is close to or even exceeds the identification level of palaeontologists (*Hsiang et al., 2019*). *Hsiang et al. (2019)* collected the accuracy of foraminiferal identification by palaeontologists and found that human accuracy is only 71.4%, which is lower than automatic identification (87.4%). Another study of planktonic
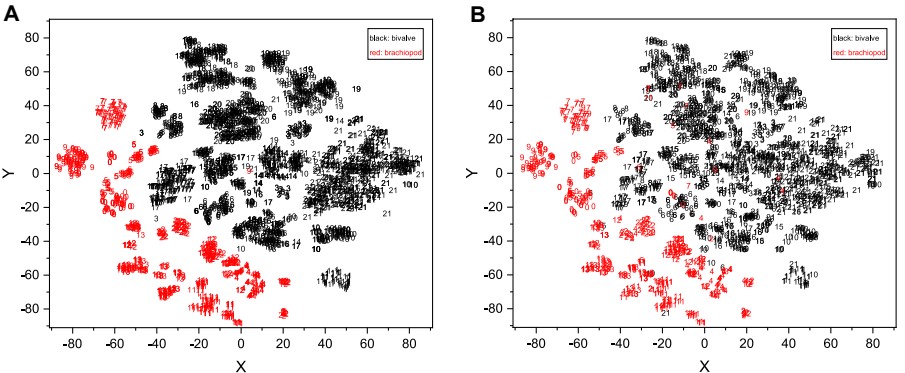

**Figure 9 Fossil morphological feature distribution maps.** (A) Training set data and (B) validation set and test set data were fitted simultaneously using t-SNE. The accuracy of the original identification model is 81.01%. The horizontal and vertical coordinates in the figure are the two dimensions obtained by t-SNE ($n$_components = 2, perplexity = 10, init = 'pca', learning_rate = 1, $n$_iter = 6,000, $n$_iter_without_progress = 6000). The numbers represent different genera, where the black numbers represent the bivalves and the red numbers represent the brachiopods. The detailed correspondence is 0, *Pseudospiriferina*; 1, *Quemocuomegalodon*; 2, *Burmirhynchia*; 3, *Promyalina*; 4, *Araxathyris*; 5, *Spiriferina*; 6, *Costatoria*; 7, *Fusichonetes*; 8, *Pteria*; 9, *Paryphella*; 10, *Neoschizodus*; 11, *Prelissorhynchia*; 12, *Juxathyris*; 13, *Piarorhynchella*; 14, *Leptochondria*; 15, *Daonella*; 16, *Unionites*; 17, *Bakevellia*; 18, *Halobia*; 19, *Eumorphotis*; 20, *Monotis*; 21, *Claraia*.

foraminifera covering 300 specimens reported an average identification accuracy of <78% for 21 experts (*Al-Sabouni et al., 2018*). In an automatic identification of modern dinoflagellates, the expert's accuracy was also only 72% (*Culverhouse et al., 2003*). *Austen et al. (2016)* found that the accuracy of experts in bumblebees was even lower than 60%. It must be noted, however, that the above-mentioned studies differ from this study in terms of the taxa and there may be differences in the difficulty of identification.

As mentioned previously, this study achieved automatic identification of fossils including 22 genera of bivalves and brachiopods, with a test set accuracy >80%. The obtained model performed relatively well considering the volume of categories and datasets in this task. *Dionisio et al. (2020)* also trained a model for identifying radiolarian fossils (containing only nine genera with 929 photographs) automatically. The accuracy of the CNN model is 91.85%, higher than ours. The average number of images per genus used in this study was comparable to ours; however, they used SEM photographs from the same source. Fewer extraneous factors and fewer categories might have contributed to slightly higher accuracy. Models for the automatic identification of pollen from 16 genera were also proposed with accuracies between 83% and 90%, also using microscopic images (*Romero et al., 2020*).

Moreover, models based on BBFID performed similarly to the models based on the corresponding scale of BBFID-1 or BBFID-2, which indicates that the ATIM is not easily affected by the similar morphology between bivalves and brachiopods with sufficient data volume (as further demonstrated by the confusion matrix). The models are highly reliable in bivalves and brachiopods identification at the genus level, which provides a basis for our subsequent comparison of their morphology. Moreover, the identification accuracy of

BBFID (scale C, including 379 taxa) was 53.71%, which is understandable considering the large taxonomic unit number with the relatively limited training set. Large-scale automatic fossil identification based on a small dataset is feasible. However, it must be noted that the categories with fewer figures are more concentrated in the literature, which might have led to the similarity between the test set and the training set. Thus, these accuracies cannot objectively generalize the performance and ability of models.

Regarding species-level automatic identification performance, we achieved an accuracy of 82.83% for 16 species identification, with several species attributed to the same genus with relatively similar morphology. Although *Kong, Punyasena & Fowlkes (2016)* automatically identified three pollen species of the same genus in a confusing species classification task with 86.13% accuracy, it must be noted that their pollen task relied more on confusing information such as a texture for identification. Importantly, the identification accuracy of mixed data scale C in the species mode is similar to, or even slightly higher than, that in the genus mode. This implies that the number of taxonomic categories can have a greater impact on automatic identification performance relative to the differences between taxonomic units. The relationship between the number of categories and the accuracy corroborates this (Appendix S4). The two correspond well to the logarithmic relationship ($R^2 = 0.8975$).

Although we independently built a dataset containing >16,000 images, it is still small for machine learning. Most studies in automatic fossil identification have focused on a few categories and large sample sizes (*Liu & Song, 2020*; *Liu et al., 2023*; *Niu & Xu, 2022*; *Wang et al., 2022*), which undoubtedly helps improve performance. *Niu & Xu (2022)* used a dataset of 34,000 graptolites to perform an automatic identification study of 41 genera, which resulted in 86% accuracy. In contrast, the identification accuracy of 47 genera in this study was 76.26%, which demonstrates the importance of larger data sets.

## Analysis of identification results

We tested models in genus mode using BBFID-1, BBFID-2, and BBFID (scale A) and obtained a confusion matrix (Figs. 4, 5 and 6), which truly reflects the model performance and misidentification. Example images of all 22 genera in this scenario are shown in the Appendix S5 for a better comparison of morphological differences. In the confusion matrix, the vertical axis represents the "true" genus name, whereas the horizontal axis represents the "predicted" genus name. The numbers in the matrix represent the proportion of "true" genera identified as "predicted" genera, and the larger the proportion, the darker the squares. The model performed well in the automatic identification of bivalves and brachiopods respectively, and misidentification was maintained at a low level.

In the hybrid auto-identification model (*i.e.,* model based on BBFID), the overall performance was good although the accuracy (81.90%) decreased slightly compared to the separate auto-identification accuracies of bivalves and brachiopods (*i.e.,* accuracy testing by BBFID-1 or BBFID-2). The genus *Quemocuomegalodon* maintained a high identification recall (1.00) in the bivalve categories, whereas the recall of *Proyalina* increased from 0.88 to 0.92. Other categories decreased slightly. Most of the brachiopod categories showed significant or stable increases, whereas only two genera exhibited recall decreases (*Araxathris*

from 0.76 to 0.68 and *Paryphella* from 0.77 to 0.72). The change in the recall may be related to the change in the distribution of the training set. Among these misidentified categories, two cases were distinctive, each exceeding 0.20 of their respective categories in the test set. The bivalve *Pteria* was misidentified as *Bakevellia* (0.25), and the brachiopod *Paryphella* was misidentified as *Fusichonetes* (0.24), with morphological similarity being the main reason for misidentification. For example, the shells of both *Pteria* and *Bakevellia* have similar outlines and are anteriorly oblique. The posterior ear is larger than the anterior ear. Distinctive concentric rings are visible on the shell surface. All these features are very similar.

Importantly, the vast majority of misidentifications in the hybrid auto-identification model occurred within categories (*i.e.,* bivalves were misidentified as other bivalves and brachiopods were misidentified as other brachiopods), whereas misidentifications between broad categories were relatively rare. For example, only 0.04 of the brachiopod *Araxathris* were misidentified as bivalve *Daonella* and 0.04 as bivalve *Eumorphotis*, which indicates that bivalves and brachiopods have considerable morphological differences.

The above are all cases where the input fossil taxon is included in the training set, but in reality, there are many fossil taxa that are not included in the training set. To deal with this exception, we propose an AM to identify such cases. The accuracy of AM (suitable for Order 22) is 85.54%. When the training is completed, the user can use the AM to verify whether the taxon of the input images is included in the training set and the usability of the genus/species identification model. If the result is "applicable", the fossil will be identified automatically. If the result is "inapplicable", the identification model will give the name of the fossil taxon that is most similar to it, and the user can continue the manual identification based on that taxon.

## Morphological analysis of fossils

Fossils have complex and variable high-dimensional morphological features, which are difficult to visualize and analyze. Deep learning can extract features, downscale dimensions of data, and exclude the influence of human bias to fully reflect the fossil features. Neural networks can extract features more efficiently than manually selected features, although the majority of the data extracted by models are too abstract for the human eye (*Keceli, Kaya & Keceli, 2017*). The accuracy of supervised classification of ammonoids using human-selected geometric features (*Foxon, 2021*) was similar to the accuracy in this study.

Machine learning can quantify morphological features and compare differences. In the feature map (Fig. 8), we can observe the identification features used by the convolutional neural network. However, the supervised deep learning used in this paper is a "result reason" approach that cannot verify the correctness of the taxonomic practice. Models may use some features not used by experts to identify, which does not mean that the taxonomic practice is wrong. A possible scenario is that there are multiple differences between the two taxa, with experts and models choosing different perspectives. The model establishes a relationship between the input (fossil image, *i.e.,* morphological features) and the output (taxon), and its ability to accurately identify fossil taxa indicates that taxonomic practice is well correlated with fossil morphology. However, the features used

by the models sometimes differ from those used by humans (*Liu et al., 2023*). Input–output relationships are established by feature extraction through convolutional neural networks. Automatic identification relies on these features that are similar with the working process of experts. The features extracted by the model are diverse, such as the umbilicus, ribs, and inner whorl of the ammonoid, spires and apices of gastropod, and growth lines and radial ribs of bivalve and brachiopod (*Liu et al., 2023*). For the identification results, there is no difference between the model's identification using images (actually fossil morphology) and the expert's identification using characterization. This is essentially determined by the prior knowledge, which is obtained by taxonomic practice. In the future, unsupervised learning may be able to provide unique insights to evaluate taxonomic practice.

In the downscaled visualization of this model for the validation and test sets, the brachiopods and bivalves are clearly demarcated, but a few points are still mixed (Fig. 9B). A clear boundary means that the brachiopod and bivalve fossils are sufficiently morphologically distinct, so that the model can extract the differences well and represent them quantitatively. This demonstrates the unique potential of deep learning models for fossil feature extraction. Without inputting any prior knowledge other than the genus name (*e.g.*, the model does not know which genus belongs to bivalve or brachiopod), the model computationally obtains information on the morphological differences between bivalve and brachiopod, which is compatible with the expert's classification. In addition to the distinction between bivalve and brachiopod, the t-SNE gives an indication of the similarity of fossil morphology. For example, in Fig. 9B numbers 8 (*Pteria*) and 17 (*Bakevellia*) overlap more, which demonstrates their more similar morphology. This is the same as the traditional morphological view. In the future, it may be possible to use this feature to find similar classification boundaries relying on models to perceive more detailed information about fossils (*e.g.*, ornamental features and 3D-morphology), which in turn could allow for quantitative differentiation of gradual features (*Klinkenbußet al., 2020*; *Edie, Collins & Jablonski, 2023*). That could not only provide new possible perspectives for exploring fossil classification and biomorphological evolution, but also try to explore whether there are important features that have been overlooked by experts. In terms of the distribution area, the distribution of bivalve points is more extensive than that of brachiopods, indicating that bivalves have greater morphological variability than brachiopods in our dataset (but the effect of image context is not excluded here). Overall, the fossil features extracted by CNNs can reflect the morphological characteristics of organisms to some extent.

CNNs can complement existing methods for morphological studies such as morphological matrix (*Dai, Korn & Song, 2021*), landmark (*Bazzi et al., 2018*), fractal dimensions (*Wiese et al., 2022*), ornamentation index (*Miao et al., 2022*), conch properties (*De Baets, 2021*), and 3D morphological methods (*Klinkenbußet al., 2020*) and provide new perspectives for studying the morphological evolution of fossils in the future. Geometric morphometry requires the extraction of fossil features by labelling manually and performing descending operations (*e.g.*, principal component analysis), which has proven to be very effective (*Aguirre et al., 2016*; *Topper et al., 2017*). In this method, fossil features are selected by experts, with biological significance and better interpretation. However, it is also influenced by human factors, and some features may be missed (*Villier & Korn, 2004*;

*Dai, Korn & Song, 2021*). Artificial intelligence differs in that it can obtain all information displayed in fossil images (not just a few dozen points). These obtained features are then downscaled (*e.g.*, t-SNE used in this paper) to get the final fossil features. However, due to the black-box character of deep learning, the features obtained are poorly interpretable, and whether they are biologically meaningful needs further study in the future. Therefore, the advantage of artificial intelligence mainly lies in the feature extraction, which reduces the subjective influence and the time cost of manual marking. On the other hand, manual feature extraction is difficult to orient to a large number of specimens and is based only on some specific species. However, deep learning is capable of obtaining information from more specimens at the scale of big data, such as intraspecific differences, spatial and temporal differences, *etc.*, due to its ability to automate the extraction of fossil features. Moreover, combining 3D information of fossils for palaeontological studies is also promising (*Hou et al., 2020*).

## CONCLUSIONS

In this study, we used machine learning to automate fossil identification based on the practical needs of palaeontological research. We built a bivalve and brachiopod fossil dataset by collecting open literature, with >16,000 "image-label" data pairs. Using these data, we compared the performance of several convolutional neural network models based on VGG-16, Inception-ResNet-v2, and EfficientNetV2s, which are commonly used in the field of image classification and fossil identification. For this identification task, we found that EfficientNetV2s has the best performance.

We finally achieved automatic fossil identification including 22 fossil genera (genus mode, based on BBFID, including 13 bivalve genera and 9 brachiopod genera) and 16 fossil species (species mode, based on BBFID, including 8 bivalve species and 8 brachiopod species), both with >80% accuracy. Furthermore, we conducted a study on the multiple categories' automatic fossil identification at the species level, and the test accuracy was ~64% based on BBFID (scale C, containing 343 bivalves and brachiopods). Models performed well in the automatic identification of multiple categories with a small dataset. These models can be deployed to a web platform (www.ai-fossil.com, *Liu et al., 2023*) in the future to make them accessible more easily and usable by researchers. At present, automatic fossil identification must be based on expert consensus, which is precisely why we emphasize the use of this model primarily for common fossil categories to aid in identification. With more taxa be included, we can use the output from deep learning models to accelerate the systematic palaeontology work during research rather than replace it and contribute to quantitative assessment of morphology (*De Baets, 2021*). Therefore, researchers can focus on the most challenging and ambiguous identification cases. When a new taxon is found, the AM output is "inapplicable", and experts can perform further taxonomic studies on it. When experts decide to establish a new species, the fossil differences given by the algorithm can assist them in making determinations, which is what the model excels at. But ultimately the establishment of new species still depends on how taxonomists apply the results of deep learning. We believe that there will be many palaeontologists working

on fossil taxonomy and creating a steady stream of a priori knowledge to promote the interdisciplinary relationship between palaeontology and computer science together with AI researchers.

However, it must be noted that the model is an exploratory experiment and can currently only serve as a useful assist to manual identification, not a complete replacement for it, at least for now. The current model still relies on a manually created taxonomy and uses it as a priori knowledge for model training. Current models are not able to combine all biological features (now only use morphological data) to build the taxonomy by themselves. However, when experts have completed the taxonomic criteria, researchers can use AI to identify fossils based on those criteria, reducing repetitive identification work and allowing palaeontologists to have more time and energy for other more creative research work.

We also used machine learning to extract high-dimensional data of fossil morphology and downscaled them to obtain fossil morphological feature distribution maps, which present the similarity of fossil morphology in a visual way. It was found that the bivalve and brachiopod distribution regions have distinctive boundaries, and the morphological differences between the two are obvious enough from the neural network perspective. In this process, models based on deep learning are not absolutely objective. In contrast, palaeontologists play a crucial role. This is precisely why we chose researcher consensus as a priori knowledge. Furthermore, we downscaled the fossil features to cast the map and observe their morphological distribution. Compared with the manually selected features, features based on the models are more objective and can better reflect the morphological characteristics of fossils, which are still derived based on the consensus of researchers on fossil taxonomy to a certain extent. In the future, this can be used as a basis to quantify morphological information, analyze their morphological spatial distribution, and provide a new perspective for exploring biological evolution.

### Funding

This study is supported by the National Natural Science Foundation of China (41821001, 42072010), 111 Project (B08030), State Key Laboratory of Biogeology and Environmental Geology (GBL22103) and the Fundamental Research Funds for the Central Universities, China University of Geosciences (Wuhan). This is the Center for Computational & Modeling Geosciences Publication Number 7. The funders had no role in study design, data collection and analysis, decision to publish, or preparation of the manuscript.

### Grant Disclosures

The following grant information was disclosed by the authors:
National Natural Science Foundation of China: 41821001, 42072010.
111 Project: B08030.
State Key Laboratory of Biogeology and Environmental Geology: GBL22103.
Fundamental Research Funds for the Central Universities, China University of Geosciences (Wuhan).

This is the Center for Computational & Modeling Geosciences Publication Number 7.

## Competing Interests

Haijun Song is an Academic Editor for PeerJ.

## Author Contributions

- Jiarui Sun conceived and designed the experiments, performed the experiments, analyzed the data, prepared figures and/or tables, authored or reviewed drafts of the article, and approved the final draft.
- Xiaokang Liu conceived and designed the experiments, performed the experiments, analyzed the data, authored or reviewed drafts of the article, and approved the final draft.
- Yunfei Huang analyzed the data, authored or reviewed drafts of the article, and approved the final draft.
- Fengyu Wang analyzed the data, authored or reviewed drafts of the article, and approved the final draft.
- Yongfang Sun analyzed the data, authored or reviewed drafts of the article, and approved the final draft.
- Jing Chen analyzed the data, authored or reviewed drafts of the article, and approved the final draft.
- Daoliang Chu analyzed the data, authored or reviewed drafts of the article, and approved the final draft.
- Haijun Song conceived and designed the experiments, analyzed the data, prepared figures and/or tables, authored or reviewed drafts of the article, and approved the final draft.

## Data Availability

The BBFID is available at Zenodo: Sun, Jiarui, Liu, Xiaokang, Huang, Yunfei, Wang, Fengyu, Sun, Yongfang, Chen, Jing, Chu, Daoliang, & Song, Haijun. (2022). Bivalve and Brachiopod Fossil Image Dataset (BBFID) [Data set]. Zenodo. https://doi.org/10.5281/zenodo.7248780.

The main code and models of this study are available at Zenodo: Sun, Jiarui, Liu, Xiaokang, Huang, Yunfei, Wang, Fengyu, Sun, Yongfang, Chen, Jing, Chu, Daoliang, & Song, Haijun. (2023). Automatic identification and morphological comparison of bivalve and brachiopod fossils based on deep learning. Zenodo. https://doi.org/10.5281/zenodo.8126697.

## Supplemental Information

Supplemental information for this article can be found online at http://dx.doi.org/10.7717/peerj.16200#supplemental-information.

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
