# Peer review of "Automatic identification and morphological comparison of bivalve and brachiopod fossils based on deep learning"

_PeerJ, doi:10.7717/peerj.16200_

## Round 0.1 · original submission · Major Revisions

I agree with the reviewers that your study is an important contribution to the community and would like to see it published not just for the massive dataset you provide but also for the comparative analyses between brachiopods and bivalves adding a novel dimension to such studies. However, there some crucial aspects which need to be addressed before publication. The main points being:

Novelty and importance: you used very similar methods on a large fossil image dataset, but it is not made clear enough how this research differs from this study (compare reviewers 1, 2 and 3). I agree with reviewer 3 that points worth highlighting is that your study contains significant more taxonomic categories while retaining a comparable level of accuracy, bearing of categories on confounded by the model, the fact that your dataset was entirely created using literature mining, etc. (see review 3 for a more detailed description). I agree with the reviewer 1 that making the differences and aim of your study clearer in the introduction would help in this endeavor. Also, the need (Why is it difficult to distinguish them?) and importance (What is the advantage when they can be pre-sorted to some degree?) to distinguish brachiopods and bivalves (e.g., your study) should be addressed more clearly (also compare reviewer 3). I am aware these are common topics addressed even in undergraduate courses (students or non-experts often confuse both groups) but not everyone might be familiar the importance and need to address these factors. It would already help to cite more literature focusing on their comparing their evolution or evolutionary dominance (I made some additional suggestions in the annotated pdf).


Reproducibility: it is crucial for the sake of scientific reproducibility to describe your analyses/experiment in greater details (compare reviewers 1, 2, 3). How much photographs came from each publication and how could publications with more photographs bias your analyses (compare reviewer 1)? What time space do the images in the time span cover and why were those selected (compare reviewer 3)? Reviewer 3 highlighted the main aspects which can and need to be addressed in greater detail.

Experimental design
In general, the experimental design is well done and the analyses are performed to a high standard. However, as stated above, the several areas in the methods lack sufficient detail for replication, or are unclearly written and may lead to confusion. Further, the lack of discussion about the time span that the images in the dataset cover is a significant issue.

Comparisons with other similar machine-learning fossil ID studies: you make comparisons with other studies but potential reasons behind differences and their comparisons seem arbitrary to some degree. This can be resolved by adding additional context (see comments by reviewer 1 and myself). I feel such comparisons could potentially also benefit from adding a table or figure comparing these studies and their confidential intervals or ranges.

Impact of adding additional metadata: I agree with reviewer 1 that the relevance of your study could be further expanded by investigating how adding additional “human-selected” features such as the localities/beds where the fossils were collected affects the accuracy.

Permissions and depicted fossil species: I greatly appreciate the addition of images of the main fossil bivalve and brachiopod data and I would like them in. However, it would be crucial to state also in the manuscript if the permissions are necessary and were obtained to reproduce them. The manuscript would also benefit from adding picture of those taxa which were most difficult to distinguish.

t-SNE (morphological) analysis: the results of the morphological analyses are interesting but should be exploited more and their relevance highlighted more (compare reviewers 2 and 3). This could be resolved by expanding the discussion on the relevance and results of the t-SNE visualization (compare reviewer 3).

Figures: I agree with reviewer 3 that the figures are in good state, but color codes could be homogenized. Adding/highlighting taxa which were most difficult to distinguish would also further increase the of Figure 1 or the t-SNE visualization and the relevance of your study more generally.

Typographical/formatting issues: there are some minor typographical, language or formatting issues which need to be resolved (see suggestions by reviewer 1, 3 and annotated pdf by myself).

Availability of code: I agree with reviewer 1 that providing the used code on a platform which provides permanent DOIs is crucial at the latest upon publication and that using comments in the code would be helpful for those who did not write the code.

Please address these and all other points raised in reviews and annotated pdfs.

I look forward to receiving your revised manuscript.

·

Basic reporting

The language throughout is written in English in a way that is clear, unambiguous, and professional. There are a few instances noted below in 'Additional Comments' where the language seemed amiss and where I have provided suggestions, but these are minor.

The literature, introduction, and background are sound (one citation suggestion below). The structure is clear, the figures are relevant, and raw data have been made available, but I have some notes on the analysis code below.

The submission is self-contained.

Experimental design

The manuscript presents original primary research within the journal scope, and the investigation is performed to a high technical standard. The methods are described sufficiently to allow replication.

Validity of the findings

The findings appear to be valid, robust, and statistically sound, and the conclusions are well stated except where direct comparisons are made to similar machine-learning fossil ID articles (see Additional Comments).

The impact and the novelty however are less clear... As cited in the present study, (some of) the same authors have published very similar analyses. For example, in https://doi.org/10.1017/pab.2022.14, the authors used three convolutional neural networks (Inception-ResNet-v2, Inception-v4, and PNASNet-5-large) on a large fossil image dataset, and launched a wonderful website to facilitate this research (https://www.ai-fossil.com/). In the present study, the authors again used three convolutional neural networks (Inception-ResNet-v2, VGG-16, and EfficientNetV2s) on a large fossil image dataset. What's not clear to me is exactly what the present effort does to build upon the concept. Perhaps this could be better justified in the Introduction, making clearly what this study adds and why it is different (besides the phyla studied).

Additional comments

ABSTRACT

Should “the artificial identification of fossils” instead be “human identification of fossils” or “manual identification of fossils” ?


INTRODUCTION

Foxon (2021; https://doi.org/10.31233/osf.io/ewkx9) is cited in the discussion, but perhaps could also be cited in the Introduction. More importantly, De Baets (2021; https://doi.org/10.24072/pci.paleo.100010) should be cited in the Introduction for arguments for using machine learning in this context.


MATERIALS AND DATA

“We obtained more than 16,000 fossil images from 188 publications” – how many images came from each publication? If, for example, you extracted 100 images from one publication, and just one image from another publication, this may have the effect of biasing the algorithm toward features in photographs from the publications with more photographs.

Should “with the 14,185 items” instead be “with 14,185 items” ?


METHODS

“To improve their generalization ability and make the model easier to train, we randomly adjusted the image (training set and validation set) brightness (within ± 0.5) and contrast (within 0 to + 10) to reduce the effect of noise” – a citation is needed here to justify this method, even if it is common.

The data are made available on Zenodo (good), but the analysis code is uploaded to GitHub. GitHub is a fine platform, but Zenodo or OSF would be a better place to upload the code, because those platforms provide permanent DOIs, whereas GitHub does not. Permanency of the code is crucial.

Also, the analysis code should be fully commented to facilitate understanding for those who didn’t write it. I see no comments. Please add comments to all of the code.

Should “Because of dataset size, model’s” instead be “Because of dataset size, the model’s” ?


RESULTS

Was permission obtained to reuse the fossil images in each Figure? This perhaps needs to be clarified in the article.


DISCUSSION

“there are quite some fossil taxa” – the wording here is slightly off, perhaps “there are many fossil taxa” ?

“The accuracy of supervised classification of ammonoids using human-selected geometric features was only 70.4%-78.1% in 11 species (Foxon 2021), lower than the accuracy of > 80% for 22 species identifications in this study.” – Comparing accuracies in a general way is fine, but the phrasing here seem unnecessarily competitive. Furthermore, your abstract more honestly reports “> 80% identification accuracy at 22 genera and ~64% accuracy at 343 species”, so the “>80%” chosen in the above sentence in the Discussion is either wrong (“species” instead of “genera”) or cherry-picked. Indeed, test accuracy was much lower in some cases according to Table 2. I see no reason why you cannot instead say “The accuracy of supervised classification of ammonoids using human-selected geometric features was similar or comparable to the accuracy in this study (Foxon 2021).” There isn’t much difference between 78.1% and 80%, and the two studies consider different phyla. It’s not a competition!

An interesting discussion point to add would be the combination of human-selected features, and deep-learning-identified features. For example, the CNNs in the present study ‘knew’ what the fossils looked like, but did not know for example where the fossils were collected. It could be that additional input features such as location of collection, which the CNN cannot determine from image data, could provide even greater accuracy in future studies.

Reviewer 2 ·

Basic reporting

Thanks for the opportunity to review this interesting work by Sun et al. This work has the potential to be an important contribution to the geosciences community. One particular contribution is the BBFID dataset which contains a large number of fossil images with their label. This can be used in future works.
However, among other minor issues, the manuscript lacks a clear and strong foundation in justifying the novelty of this work and also how their t-SNE (morphological) analysis is useful. Overall, I think a major revision is required before the manuscript can be considered for publication in PeerJ.
Please go over the annotated PDF file for my detailed comments.

Experimental design

No comment

Validity of the findings

No comment

Additional comments

Please find the annotated PDF file.

Annotated reviews are not available for download in order to protect the identity of reviewers who chose to remain anonymous.

Reviewer 3 ·

Basic reporting

There are issues with the language which affects readability and comprehension. Several areas need additional details for clarity and/or reproducibility. More details on the specific areas, and how to improve them, are provided in the "Additional comments" section.

Experimental design

In general, the experimental design is well done and the analyses are performed to a high standard. However, as stated above, the several areas in the methods lack sufficient detail for replication, or are unclearly written and may lead to confusion. Further, the lack of discussion about the time span that the images in the dataset cover is a significant issue. Details on these comments can be found in the "Additional comments" section.

Validity of the findings

The findings are valid and well-stated overall, but there are some discussion topics that should be expanded in light of the authors' stated intentions/motives for building this model (e.g., expanding the discussion on the results of the t-SNE visualization). More details are found in the "Additional comments" section.

Additional comments

In their manuscript, "Automatic identification and morphological comparison of bivalve and brachiopod fossils based on deep learning", Sun et al. train a supervised machine learning model to identify genera/species of fossil bivalves and brachiopods. Their best-performing model uses EfficientNetV2's architecture and achieves >80% accuracy for 22 genera and ~64% accuracy for 343 species. Compared to previous work using deep learning to identify bivalves and brachiopods, this study contains significant more taxonomic categories while retaining a comparable level of accuracy.

The study is well-designed and touches on many important and relatively little-explored aspects of applying CNNs to automatic taxonomic identification. For instance, the comparison of t-SNE patterns between brachiopods and bivalves, as well as the discussion of which categories are confounded by the model, are useful and interesting. I also find the statement on lines 321-322 about how the number of categories is more important than the relative difference between taxonomic units for model performance very interesting. The figures are also well-designed and appropriate (with a few minor issues that I list below in the Minor Comments section below). Also, the fact that the dataset was created entirely using literature mining is quite unique, and I believe the authors would be well served by making more of this aspect and expanding their discussion about this specifically (see Major Comment #3, below).

However, overall the manuscript is hampered by a lack of information and detail in several areas, and also by a somewhat disorganized writing style that can be unclear, which negatively impacts readability and understanding. In the Major Comments section below, I have listed places where I believe more information and/or clarification is necessary. This includes some sections where more evidence is necessary to justify the claims being made. Further, the lack of discussion about the time period(s) that the dataset covers and how this affects the contained species (see Major Comment #1a below) is a significant issue.


Major Comments

1. There are several places where more information/clarification is necessary:
1a. In the introduction, it would be helpful to get more biological/geological context for the general PeerJ audience. For example, how many genera/species of bivalves and brachiopods are there (that we know of)? This would help the reader assess how representative the BBFID is. Also, what time period does this dataset cover? Brachiopods and bivalves are over 500 million years old and the number/composition of genera have changed markedly over time (see Hsieh et al. [2019] Paleobiology, for example). Does this model contain, for instance, species from time periods where the taxonomic composition does not overlap? There is no practical need to train a model to distinguish between species that are separated by million of years because they would never be considered together. Also, there can also be cases where two species are very morphologically similar due to convergent evolution, but they would not be confused by taxonomists because they occur in sediments of different ages. I'm not saying that the authors necessarily need to explicitly test all these questions in their experimental setup, but it is necessary for them to state the time period covered so that the appropriate context can be considered.
1b. Lines 169-172: What exactly is meant by "the images were normalized and standardized"? This statement does not mean much by itself. Was histogram equalization performed? Were pixel intensity ranges made the same for all images? Please specify exactly what was done. Also, why was resizing and randomized brightness/contrast adjustment done in preprocessing rather rather than using the built-in resizing and augmentation methods available in Tensorflow? I don't think it's necessarily a problem, but it is a deviation from standard practice and I'm curious why this was done.
1c. Lines 193-194: Explain what EarlyStopping is. The average reader will not know what is meant by this.
1d. Line 232: Please cite the open-source projects that the code was based on.
1e. Lines 238-240: It would be useful here to state how many images there were for each of these categories so that readers can see how the accuracy values compare with the sample sizes.
1f. Lines 269-270: "The fossil images used in this study contain pictures of the whole shells and detailed pictures, such as structures of fossils. The identification accuracy was adversely affected by this factor." The second sentence here is given no further context. How was the accuracy affected by this factor? This relates to comment #3, below -- as the authors give no indication of how many whole shell vs. detail images there are, we have no idea how to properly interpret this statement.
1g. Please explain what precision and recall mean in this context in the the Methods section before you start talking about them in the results/discussion.
1h. Lines 361-370: The description of the "Applicability Model" needs to be improved. The authors state that they divide the dataset into "applicable" and "unapplicable" categories, but do not explicitly state what these categories mean. I think what the authors mean that anything that is in the training set is considered "applicable" and anything that is not is considered "unapplicable" -- is this correct? Please clarify what is meant here to avoid confusion. Also, this model should be included in the Methods/Results sections, and not only in the Discussion section. (Also, the term used here should be "inapplicable", not "unapplicable".)

2. Some statements are not properly justified or contradictory:
2a. Lines 58-59: The authors state that "Neural network in fossil identification is still at an early stage of development and cannot yet fully reach the identification level of professional palaeontologists". Depending on how you interpret this statement, I don't believe this is correct. As the authors themselves point out, in previous studies (e.g., lines 285-291), the accuracy of deep learning models exceeds the accuracies of human experts significantly. If the authors mean that professional palaeontologists have advantages that such models do not – the ability to take into account complex contextual information, for instance – then I would be more amenable to this statement, but as it stands it is inaccurate.
2b. Line 153: "The convolutional layer reduces the data size" I'm not sure what the authors mean by this statement. Do they mean that the size of a single tensor decreases with a convolution? While that is true, it could be argued that convolution increases the data size as generally multiple filters are used in each convolution layer, which increases the dimensionality of the tensor. I would recommend that the authors remove this line as it is potentially confusing and replace it with a statement about how convolutions actually work in the context of CNNs (i.e., that they transform an image by sweeping a kernel over each pixel and performing a mathematical operation).

3. In my opinion, one of the most interesting aspects of this paper is that the dataset is generated completely from previously published work via literature mining. This is fairly unique among studies looking at applying deep learning to fossils at the moment, and as such I think the authors are missing out on an opportunity to discuss the advantages (e.g., taking advantage of the wealth of data that already exists) and consequences (e.g., account for different imaging practices across sources) of this practice specifically. While the authors do mention some of these things in the Materials and Data section and elsewhere, there are not many details about the characteristics of the dataset. For instance, are all the images grayscale? While all the example images given in Fig. 1 are grayscale, one of the input images in Fig. 7 is in color. What proportion of images had plain white/black backgrounds and what proportion were fossils in situ (i.e., within the original matrix)? What proportion of images were whole shells vs. detailed images of shell structure etc.? The lack of this information hampers the interpretation of the results presented in the manuscript.

4. On line 407, the authors mention that the results of the t-SNE visualization show that, while brachiopods and bivalves are mostly separated, there are some areas/individuals that overlap. It would be interesting and useful here to state exactly which species overlap in the space and whether these are species that are similar morphologically and/or would give human classifiers trouble as well. This is especially true because one of the main stated purposes of developing this model in the first place is to help taxonomists distinguish between brachiopods and bivalves in the fossil record.


Minor Comments

- Please add a color scale bar to figures 3, 4, 5, and appendix S1.
- In figure 5, it would be easier to visually distinguish brachiopods and bivalves on the y-axis by using either different colors or a small image/icon (rather than underlining).
- In figure 8, please show a legend in the figure itself for orange = bivalves and blue = brachiopods (i.e., don't only state this in the caption).
- I would consider using different colors for the confusion matrices. Throughput the paper you use orange and blue to designate bivalves and brachiopods, respectively, so also using orange and blue in the confusion matrices could lead to semantic wire-crossing. (Alternatively, keep the confusion matrices as they are and change the colors you use to designate bivalves/brachiopods to other ones, which is probably easier.)

---

## Round 0.2 · Minor Revisions

Many thanks for addressing our suggestions so meticulously. I agree with the reviewers that the revisions have made the manuscript easier to follow and of even broader relevance. Your manuscript is close to acceptance pending some minor, but crucial points are addressed before publication:

1) Balance of images used from different publications: I agree with reviewer 1 that the significantly larger number of images coming from 3 publications does not necessarily compromise your study but that the impact of (im)balance is crucial to discuss and why you think it is less problematic or can be tested and considered in future work. Ideally, it should be backed up by other studies who tested the impact of imbalance as well as at least discussed as a future perspective / potential limitation (compare reviewer 1)

2) Percentages of Permian and Triassic data: please explicitly state the percentages of Permian and Triassic images as well as their summed percentage on lines 129-131 when stating they represent the majority (see annotated pdf)

3) Formatting and language issues: please address the remaining minor formatting, typographical and language issues which have been nicely summarized by reviewer 3.

4) Reference list: The reference “De Baets” is misspelled in the reference list as “Baets” on Line 601 (see also annotated pdf)

I look forward to receiving the revised manuscript and seeing this work published.

·

Basic reporting

The minor wording suggestions I made in my previous review were addressed fully.

Experimental design

No change from previous review.

Validity of the findings

The authors appropriately addressed my comment on the impact/relevance of the work.

Additional comments

In general the authors have addressed my peer-review comments. The article is improved and I am satisfied with these changes in all but one instance. Sorry for digging my heels in, but I still feel that there is something left to be said about the 'balance' of images from each source, i.e., how many images came from each publication. The authors in their revision provided a beautiful figure (Appendix S2) on the contribution of each publication to the dataset, but go on to state in the article that "The contribution of publications is sufficiently balanced for the training model". It is clear from this new figure that some articles (e.g. the three articles on the left of the figure with 1002, 1254, and 1257 images) are contributing massively more toward the dataset than those which contributed <100 images each. I think that a citation is very much needed to argue that these are "sufficiently balanced", or better yet, delete that sentence ("The contribution of publications is sufficiently balanced for the training model") and instead add a sentence to the Limitations section of the article openly and honestly expressing that there was some imbalance in the contribution of each article to the dataset (citing Appendix S2) but that this was a limitation of the data sources and that future studies should explore the potential impact of this imbalance on the findings. I don't think that this compromises the study, but I do think that transparency is needed here. The article overall is improved, but I would like to see this one last change made, please and thank you. The Editor can supervise this change and the article probably does not need to go back to me.

Reviewer 3 ·

Basic reporting

There are some minor language issues throughout the manuscript -- I have listed the ones I noticed below.

Line 25-27: "An available...taxonomic determination completed)" -- This is not a complete sentence.

Line 83: "because" not "beacuse" (but I would change this word to "and" because I don't think paleontologists being able to take context into account is the most important reason for why neural networks are still in an early stage of development; rather, I would argue this is because of the large amount of time/labor/expertise/material that's required to create high-quality training datasets).

Line 85: "fewer and fewer", not "less and less"

Line 87: "as more" not "asmore"

Line 90: I'd change "In general identification field" to something like "In non-specialist identification tasks".

Line 150: I think you should break for a new paragraph starting with "Automatic identification of brachiopods has been carried out previously."

Line 175: Capitalize BMP, JPG, PNG (and throughout the manuscript).

Line 298: "used" instead of "conducted the"

Line 326-238: I would change the beginning of the sentence to "If the decrease in 'validation loss' is less than 0.0001 for 5 epochs, the learning rate..."

Line 477: "...stems from the decrease in single taxon images" instead of "...stems from the single taxon images decrease"

Line 597: Move the (Foxon 2021) citation to after "human-selected geometric features" on line 596.

Line 714: "inapplicable" not "unapplicable".

Experimental design

The authors have adequately addressed the issues I raised in my original review about reproducibility and missing methodological details.

Validity of the findings

No comment.

Additional comments

The manuscript has been much improved after revisions. After the minor language issues are addressed, I would recommend the manuscript for publication.

---

## Round 0.3 · accepted · Accept

Thank you for making this final changes which have made the manuscript even easier to follow and broader relevance. I hereby accept the manuscript pending you make verify and correct the percentages to make sure the sum of percentages listed on lines 129-131 sum to 100% (not 102.5%) during proofing phase. If Permian-Triassic is 99% together – the sum of Carboniferous, Jurassic and Quaternary cannot be 3.5% (feel free to use one decimal behind point in all cases).